# Serotonin improves behavioral contrast sensitivity of freely moving rats

**Akinori Y. Sato[1], Keisuke Tsunoda[1], Ryo Mizuyama[1], Satoshi Shimegi**[1,2,3]*

**1** Laboratory of Brain Information Science in Sports, Graduate School of Frontier Biosciences, Osaka University, Toyonaka, Osaka, Japan, **2** Laboratory of Brain Information Science in Sports, Center for Education in Liberal Arts and Sciences, Osaka University, Toyonaka, Osaka, Japan, **3** Laboratory of Brain Information Science in Sports, Graduate School of Medicine, Osaka University, Toyonaka, Osaka, Japan

* shimegi@celas.osaka-u.ac.jp

**Data Availability Statement:** All raw data files are available from the figshare (DOI: 10.6084/m9.figshare.11845611).

**Funding:** This work was supported by KAKENHI JP22500573, JP25282216, JP25560302, and

## Abstract

Serotonin (5-HT) is a neuromodulator secreted from serotonergic neurons located in the pons and upper brain stem in a behavioral context-dependent manner. The serotonergic axon terminals innervate almost the whole brain, causing modulatory actions on various brain functions including vision. Our previous study demonstrated the visual responses of neurons in the primary visual cortex (V1) of anesthetized monkeys were modulated by the activation of 5-HT receptors depending on the response magnitude, in which 5-HT2A receptor-selective agonists enhanced weak visual responses but not strong responses. This observation suggests that the activation of serotonergic receptors modulates neuronal visual information processing to improve the behavioral detectability of a stimulus. However, it remains unknown if 5-HT improves visual detectability at the behavioral level. To investigate this point, visual detectability was measured as contrast sensitivity (CS) in freely moving rats using a two-alternative forced-choice visual detection task (2AFC-VDT) combined with the staircase method. The grating contrast was decreased or increased step by step after a correct choice (hit) or incorrect choice (miss), respectively. CS was evaluated as an inverse value of the visual contrast threshold. The effect of the intraperitoneal administration of fluoxetine (FLX, 5 mg/kg), a selective serotonin reuptake inhibitor, on CS was tested. The CS of rats was significantly higher in FLX than control conditions, and the drug effect showed specificity for the spatial frequency (SF) of a grating stimulus, in which CS improvement was observed at optimal SF but not non-optimal high SF. Thus, we conclude that endogenously-secreted serotonin in the brain improves visual detectability, which may be mediated by vision-related neurons acquiring SF information of the visual stimulus.

## Introduction

Information processing in the brain dynamically changes depending on the arousal state and behavioral context [1,2]. The dynamics are mediated by neuromodulators, including 5-hydroxytryptamine (5-HT, or serotonin) depending on behavioral contexts.

Serotonin is secreted from afferent terminals originating from neurons in the dorsal and median raphe nuclei, which act on subcortical and cortical areas [3,4] to regulate various brain functions [5–7]. For example, serotonergic neurons regularly fire at low frequencies (0.2–4

JP16H01869 to S.S, and Grant-in-Aid for JSPS
Fellows JP17J08499 to A.Y.S.

**Competing interests:** The authors have declared
that no competing interests exist.

Hz) during awakening [8,9] to maintain the arousal state by acting on the hypothalamus [10,11] and cerebral cortex [12,13]. Serotonergic neurons project to the cerebral cortex including visual areas [14–16] and are known to modulate neuronal visual responses via various kinds of 5-HT receptor subtypes [17–20]. For example, Waterhouse et al. (1990) reported that iontophoretically-administered serotonin decreases the signal-to-noise ratio of the neuronal activities in the primary visual cortex (V1) of anesthetized rats [17]. On the other hand, serotonin receptor-selective agonists have bi-directional effects of facilitation and inhibition on the visual response of monkey V1 neurons depending on the response magnitude, an effect considered to optimize visual information processing [18,20].

However, since all previous studies on serotonergic visual modulation have focused on neuronal activities, it still remains unclear whether and how endogenous serotonin modulates visual perception. To examine this point, here we measured contrast sensitivity (CS) as an index of behavioral visual detectability of freely moving rats using a two-alternative forced choice-visual detection task (2AFC-VDT) combined with the staircase method, and tested the effect of fluoxetine (FLX), a selective serotonin reuptake inhibitor, on CS.

## Materials and methods

### Ethical approval

All experimental protocols were approved by the Research Ethics Committee of Osaka University (Permit Number: 28-074-000). All procedures were carried out in compliance with the policies and regulations of the guidelines approved by the Animal Care Committee of the Osaka University Medical School and National Institutes of Health guidelines for the care of experimental animals.

### Animals

Long-Evans rats (male, 200–350 g; Japan SLC Inc., Shizuoka, Japan) were used in this study. The rats were kept on a 12/12-hour light-dark cycle (lights on at 6:00 AM), and all experiments including the 2AFC-VDT and training to learn the task were performed only during the light period. Each week, five days and the remaining two days were assigned to the training/test period and non-task period, respectively. Rats obtained water only by performing the 2AFC-VDT during the training/test period, but were able to access water without any restriction during the non-task period. Signs of possible dehydration were monitored (reduced skin tension, sunken eyes, etc.), but none were observed in this study.

### 2AFC-VDT

An experimental box was used to measure CS. The box was partitioned into three areas by translucent walls, and a liquid crystal display monitor (mean luminance: 30 cd/m$^2$, maximum luminance: 60 cd/m$^2$, refresh rate: 60 Hz, screen resolution: 1920×1080 pixels, screen size: 50.9×28.6 cm) was attached to the front side (Fig 1A and 1B). Three task-related levers were equipped at the center of each area. The trial was initiated by the rat pulling the central lever. In response, a visual stimulus was presented on either the left or right side of the monitor. The stimulus was a circular patch of drifting sinusoidal grating with horizontal orientation, temporal frequency (TF) 4 Hz, and diameter 20 degrees in visual angle. When the rat pulled the lever on the side where the visual stimulus was presented (correct choice), 4–5 μL of water was given from the tip of the lever as a reward. On the other hand, when the rat pulled the opposite lever (incorrect choice), no reward and an audible sound (200–500 Hz) as an error signal were given. The visual stimulus presentation continued until the rat pulled the choice lever, and the

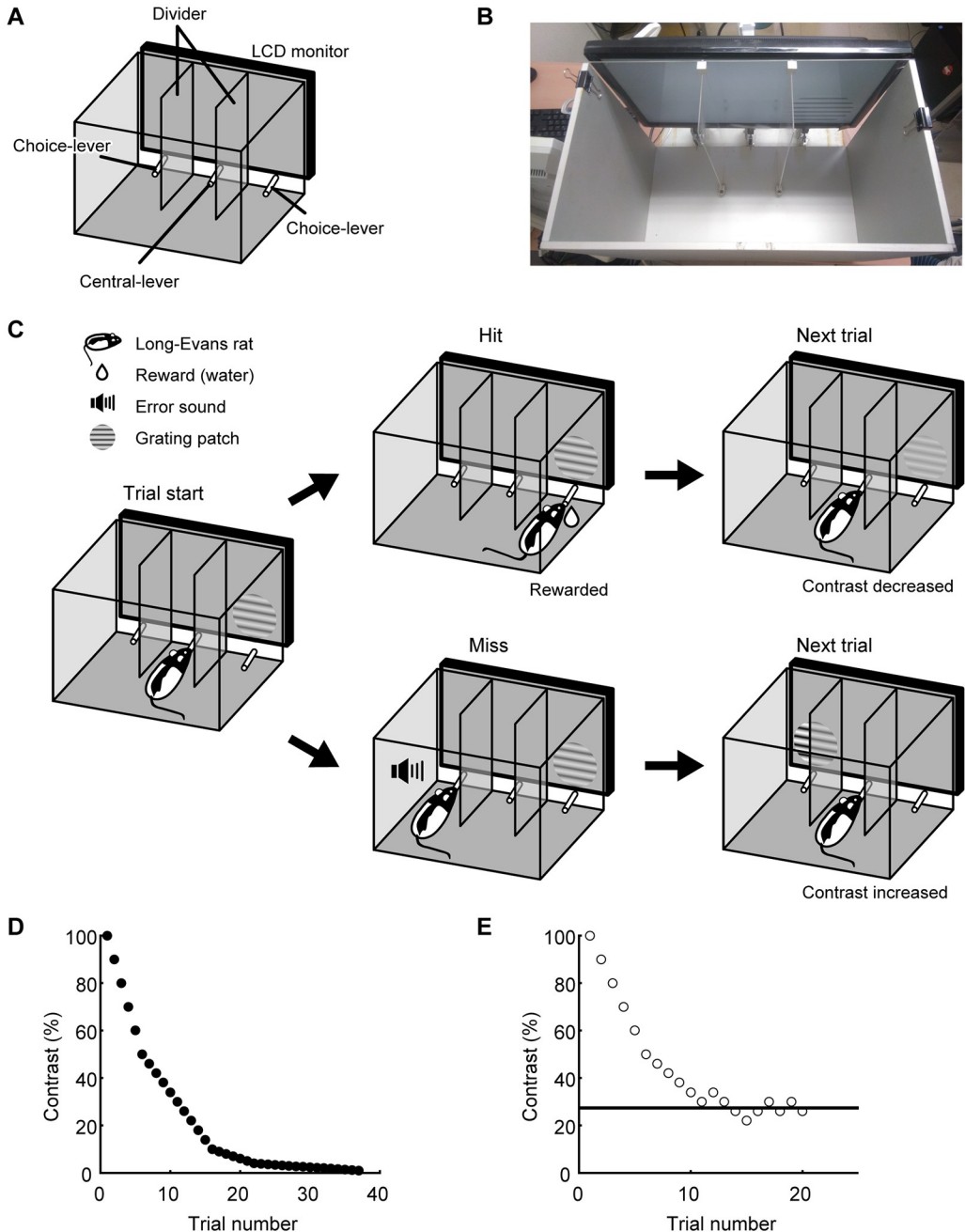

**Fig 1. Schema of the CS measurement using 2AFC-VDT combined with the staircase method.** (A) Schema and (B) photo of the experimental box. (C) Schema of the 2AFC-VDT. A rat pulls up the central lever to begin the task, by which a grating patch is presented on the right or left side of the display. The rat is rewarded with water by pulling the lever corresponding to the stimulus location. (D) The contrast changes in one session if the rat pulls the correct lever in all trials. (E) Typical result from one session in the 2AFC-VDT combined with the staircase method. In this session, the rat was able to detect 27.3% contrast (horizontal line, $C_{threshold} = 27.3\%$), therefore CS was assessed as 3.66.

stimulus location was pseudo-randomly changed from trial to trial. The rat's behavior during the task was monitored and recorded by a web camera. Details of the training protocol of the 2AFC-VDT have been described in our previous study [21]. All systems were controlled by programs written in MATLAB (Mathworks, Natick, MA) with extensions from the Psychophysics Toolbox [22,23].

Animal age at the beginning of the task training ranged from 70 to 100 days old, and the task training was mostly completed within 2 weeks. Experiments for the CS measurements were conducted over the subsequent 2–3 months.

## CS measurements

To measure CS, 2AFC-VDT combined with the staircase method was conducted [24–27]. In the staircase method, the contrast of the grating stimulus was changed according to the rat's choice in the prior trial. Upon pulling the correct choice-lever (hit) or incorrect one (miss), the grating contrast was decreased or increased, respectively. In one session of the CS measurement, the contrast changed at increments of 10%, 4%, 1%, or 0.2% for high (50–100%), middle (10–50%), low (4–10%), or very low (1–4%) contrast ranges, respectively (Fig 1D). The session was finished when the average hit rate in the last 10 trials reached 50% (chance level) (Fig 1E), and the average stimulus contrast in the last 3 trials of the session was regarded as the contrast threshold ($C_{threshold}$) of the rat. CS was calculated as CS = $100/C_{threshold}$ [28].

One session was repeated for 1 hour, and the geometric average of the CS obtained from all sessions was determined as the CS of the rat. CS depends on the spatial frequency (SF) of the stimulus, with the highest CS occurring at optimal SF in rats as well as in humans [24]. Therefore, there is a possibility that the effect of serotonin on CS is SF-dependent. To confirm the possibility, we conducted CS measurements at optimal SF (0.1 cycle/degree (cpd)) and non-optimal high SF (0.5 cpd).

## Open field test

The open field test was used to evaluate the locomotor activity of freely moving rats. In this test, a 60-cm diameter circular container was used as the open field. The movement of the rat was monitored by a web camera from above for 5 min. The total moving distance in the 5 min was calculated by using a custom-made tracking system and evaluated as the locomotor activity.

## Water intake measurement

To examine the motivation for drinking water, water intake was measured. Rats were put in their home cage and allowed to access the water bottle for 5 min. The weight of the bottle was measured before and after the 5 min period, and the difference was evaluated as the water intake of the rat.

## Drugs

Fluoxetine hydrochloride (FLX, a selective serotonin reuptake inhibitor dissolved in 0.9% saline; 5 mg/kg i.p.; Sigma Aldrich, MO) was injected intraperitoneally 30 min before the 2AFC-VDT, open field test and water intake measurement. The dose of the drug was determined on the basis of a previous study [29].

## Statistical analyses

The normally distributed data obtained from the 2AFC-VDT were analyzed using a two-way analysis of variance (ANOVA) for repeated measures. The factors were drug (control, FLX) and SF (0.1, 0.5 cpd). When the ANOVA showed a significant interaction between drug and SF, post hoc analysis using the Holm–Bonferroni method for multiple comparisons was performed. Statistical analyses of other data obtained from the open field test and water intake measurement were carried out using the paired t-test. These statistic tests were performed

using MATLAB and R version 3.5.2 (The R Foundation for Statistical Computing, Vienna, Austria). Significance criterion (alpha) was 0.05 (two-tailed). Data in the text and figures are mean ± SEM.

## Results

To investigate the modulatory effects of serotonin on behavioral visual function, we measured CS in freely moving rats using the 2AFC-VDT and tested the effects of intraperitoneally-administered FLX.

Fig 2 shows typical results of the CS measurement. In one session, as the trial number progressed, the stimulus contrast decreased step-by-step and reached a certain level ($C_{threshold}$) for any SF or drug condition (Fig 2A and 2B). However, the drug effect on $C_{threshold}$ differed between the two SF conditions (0.1 and 0.5 cpd). Under optimal SF condition (0.1 cpd), FLX

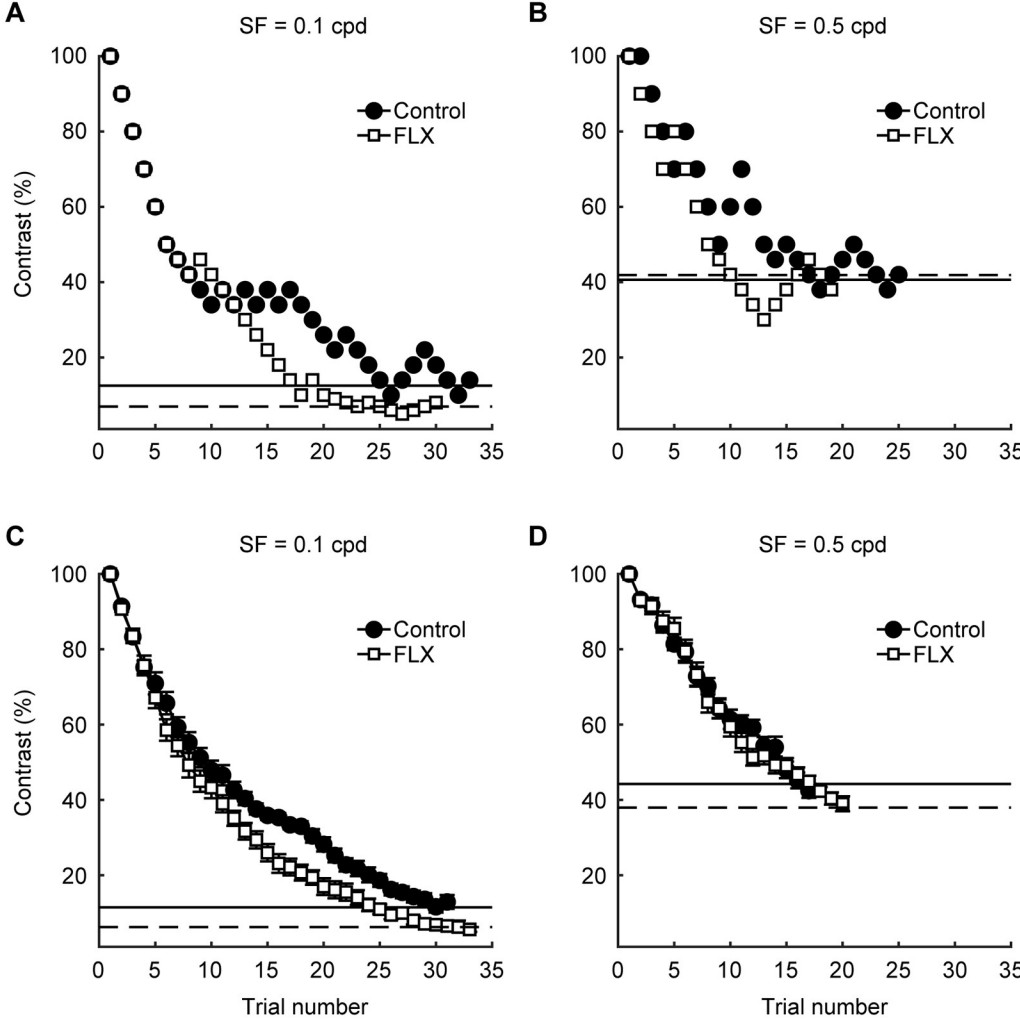

**Fig 2. Effects of FLX on CS in the 2AFC-VDT combined with the staircase method.** (A)–(B) Typical data from one session of the 2AFC-VDT at two different SFs: 0.1 cpd (A) and 0.5 cpd (B) for one rat (black circles, control; white squares, FLX). Solid and dotted horizontal lines indicate the $C_{threshold}$ of the control and FLX conditions, respectively. (C)–(D) Average contrast changes over all sessions for the same rat as (A)–(B). Black circles and white squares indicate the mean contrast value calculated for each trial number. Horizontal lines indicate the geometric mean of $C_{threshold}$ (solid line, control; dotted line, FLX). Error bars are SEM.

decreased $C_{threshold}$ from 12.5% to 7.0% and increased CS from 8.0 to 14.4 (Fig 2A). On the other hand, the CS did not differ between the two drug conditions at non-optimal high SF (0.5 cpd), in which the $C_{threshold}$ values were 40.6% and 41.9% for control and FLX conditions, respectively, and the corresponding CS values were 2.5 and 2.4 (Fig 2B). The averaged results over all sessions of the same rat showed similar results (Fig 2C and 2D). The contrast values of the stimulus for all 60 min sessions were averaged over all trial numbers, and the averaged data were plotted as average transition curves (Fig 2C and 2D). Again, FLX lowered $C_{threshold}$ at optimal SF (control: 11.5%, FLX: 6.3%), but not at non-optimal high SF (control: 44.2%, FLX: 38%). CS at optimal and non-optimal high SF were 8.7 and 2.3, respectively, for the control condition, and 16.0 and 2.6 for the FLX condition.

The results of the population analysis of the CS for all rats (n = 7) reveal the effects of FLX on CS are SF-dependent (Fig 3). A two-way ANOVA for repeated measures was used for the data analysis, and statistical significance was observed for the treatments ($F_{(1, 24)}$ = 14.13, $p$ = 0.004), SFs ($F_{(1, 24)}$ = 20.18, $p$ = 0.009) and the interaction ($F_{(1, 24)}$ = 6.439, $p$ = 0.044). The post hoc analysis using the Holm–Bonferroni's multiple comparison test showed that FLX improved CS significantly only at optimal SF (SF 0.1 cpd, $p$ = 0.022), but not at non-optimal high SF (SF 0.5 cpd, $p$ = 1). These results suggest that increasing the serotonin concentration in the brain improves behavioral sensitivity to stimulus contrast in an SF-dependent manner.

Serotonin in the brain elevates the motivation for drinking water and changes the activity level of the animal [30,31]. Changes in drinking motivation and activity level could affect performance in the 2AFC-VDT to improve CS. To investigate this point, water intake and activity

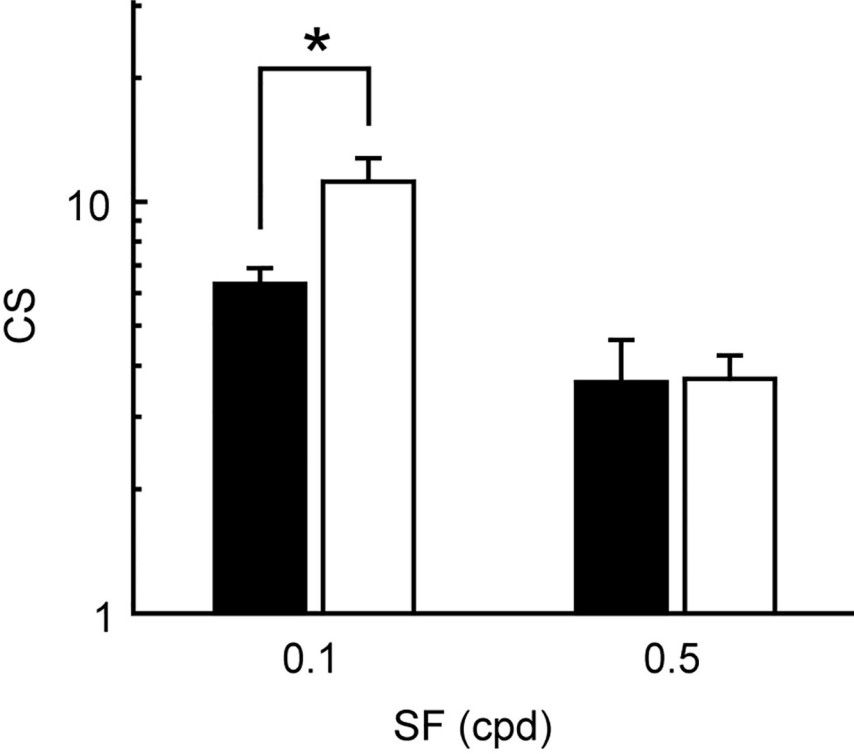

**Fig 3. Population data of FLX effects on CS.** The population data of CS (n = 7) obtained from control and FLX conditions (black bar, control; white bar, FLX). CS was significantly increased by FLX at 0.1 cpd (*$p$ = 0.022, Holm–Bonferroni's multiple comparison test), but not at 0.5 cpd. Error bars are SEM. The vertical axis is shown in logarithmic scale.

level for 5 min were measured after the administration of saline or FLX (Fig 4), but we found no difference between the two conditions (water intake: control, 5.1 ± 0.7 g, FLX, 4.5 ± 0.7 g, $p$ = 0.369; average moving distance: control, 24.0 ± 1.25 m, FLX, 22.3 ± 1.75 m, $p$ = 0.570, paired t-test).

## Discussion

In the present study, we examined the effects of FLX, a serotonin-selective reuptake inhibitor, on the CS of freely moving rats, finding that CS was enhanced only at optimal SF. This finding suggests that increasing serotonin in the brain improves behavioral visual detectability in an SF-dependent manner.

The SF dependency of the FLX effect suggests that the candidate target area of FLX is the visual areas, which represent the SF of a visual stimulus, for example, area V1 [32]. Our previous neuropharmacological studies [18,33] demonstrated that serotonin affects the contrast-response function of neurons in monkey V1, where the 5-HT receptor subtypes, 1B and 2A are present exclusively among neocortical areas. Interestingly, the iontophoretic administration of DOI, a 2A receptor agonist, directly to the recorded neuron caused bi-directional modulation on the contrast-response function, in which weak responses at low contrast were enhanced but strong responses at high contrast were suppressed [18,33]. DOI's effect at low contrast is consistent with our finding that FLX enhances CS, that is, reduces the $C_{threshold}$ at which rats become unable to detect a grating stimulus. The activation of 5-HT 2A receptor has been reported to cause bi-directional modulation in rat V1 neurons depending on the magnitude of the visual responses, in which weak visual responses were facilitated but strong ones

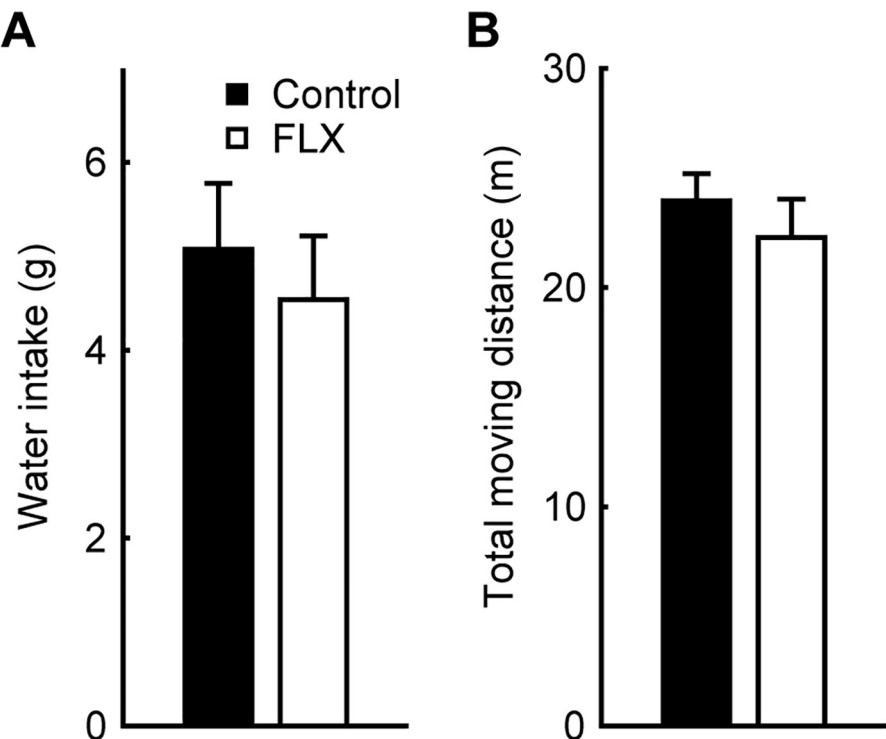

**Fig 4. Effect of FLX on water intake and activity level.** (A) Water intake for 5 min under control and FLX conditions (black bar, control; white bar, FLX). No significant difference was observed ($p$ = 0.369, paired t-test). (B) Total moving distance of rats in an open field arena for 5 min was also unaffected by FLX administration ($p$ = 0.570, paired t-test).

suppressed [20]. Thus, 5-HT 2A receptors seem to play an important role in the enhancement of weak visual signals, potentially improving the behavioral CS observed in the present study. On the other hand, 1B receptor is reported to be located on thalamocortical axons projecting to area V1 of rats [34], and its activation reduces spontaneous activity, i.e., noise [33], which would enhance the signal-to-noise ratio responsible for behavioral CS. Further study is required to elucidate whether and how 5-HT receptors in area V1 contribute to CS improvement.

Serotonin is released from serotonergic neurons in the dorsal and median raphe nuclei. The neurons of both regions project their axons to the occipital cortex, but density of the projection is histologically known to be higher in dorsal raphe nucleus than median raphe nucleus [35–37]. Therefore, comparing to the median raphe nucleus, the dorsal raphe nucleus seems to contribute more to the modulatory effect of serotonin in the visual cortex.

In the present study, FLX was intraperitoneally administered to minimize invasiveness, but this strategy has the disadvantage of wide-ranging effects across the brain areas where serotonergic neurons project to. To determine if brain areas other than visual areas and peripheral nervous system were influenced by this administration, we examined water intake and activity levels. Serotonin has been known to induce intake behavior toward food and water via the accumbens nucleus [31]. However, we found FLX did not change water consumption volume. Additionally, we found FLX did not change activity level.

In addition, it has been reported that FLX not only inhibits serotonin transporters but also acts as a 5-HT 2B receptor selective agonist [38]. Therefore, FLX's modulatory effect on CS may be mediated through activation of 5-HT 2B receptors as well as increment of serotonin concentration. In cerebral cortex of adult rat brain, the expression of the 5-HT 2B receptor mRNA is not observed [39–41]. On the other hand, the immunoreactivity of 5-HT 2B receptors protein was reported to be positive in frontal cortex but not in other cortical areas [42]. Therefore, it is considered that the contribution of activation of 5-HT 2B receptor to the CS improvement by FLX administration is small in this study.

In conclusion, serotonin endogenously released from the dorsal and median raphe nuclei during the 2AFC-VDT can improve CS depending on the SF of the visual stimulus, in which visual detectability at the most sensitive SF (optimal SF) which is considered to carrier the most valuable and available visual information for behavior, is significantly enhanced.

## Acknowledgments

We thank Dr. Hiromichi Sato for discussions and comments, and thank Dr. Peter Karagiannis for improving the English of the manuscript. A.Y.S. and S.S. designed the research, A.Y.S. collected the data and performed the analyses, A.Y.S., K.T., and R.M. provided technical assistance for the task training and CS measurement, and A.Y.S. and S.S. wrote the paper.

## Author Contributions

**Conceptualization:** Akinori Y. Sato, Satoshi Shimegi.

**Data curation:** Akinori Y. Sato.

**Formal analysis:** Akinori Y. Sato.

**Funding acquisition:** Akinori Y. Sato, Satoshi Shimegi.

**Investigation:** Akinori Y. Sato, Satoshi Shimegi.

**Methodology:** Akinori Y. Sato, Keisuke Tsunoda, Ryo Mizuyama, Satoshi Shimegi.

**Project administration:** Satoshi Shimegi.

**Resources:** Satoshi Shimegi.

**Software:** Akinori Y. Sato, Keisuke Tsunoda, Ryo Mizuyama.

**Supervision:** Satoshi Shimegi.

**Validation:** Akinori Y. Sato, Satoshi Shimegi.

**Visualization:** Akinori Y. Sato.

**Writing – original draft:** Akinori Y. Sato, Satoshi Shimegi.

**Writing – review & editing:** Akinori Y. Sato, Satoshi Shimegi.

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
