## [Decision Letter · Decision Letter 0]

3 Jan 2020

PONE-D-19-31125

Serotonin improves behavioral contrast sensitivity of freely moving rats

PLOS ONE

Dear Dr. Shimegi,

Thank you for submitting your manuscript to PLOS ONE. After careful consideration, we feel that it has merit but does not fully meet PLOS ONE’s publication criteria as it currently stands. Therefore, we invite you to submit a revised version of the manuscript that addresses the points raised during the review process. These comments can be found below.

We would appreciate receiving your revised manuscript by Feb 17 2020 11:59PM. To enhance the reproducibility of your results, we recommend that if applicable you deposit your laboratory protocols in protocols.io, where a protocol can be assigned its own identifier (DOI) such that it can be cited independently in the future. For instructions see: http://journals.plos.org/plosone/s/submission-guidelines#loc-laboratory-protocols

We look forward to receiving your revised manuscript.

Kind regards,

Judith Homberg

Academic Editor

PLOS ONE

Journal Requirements:

"This work was supported by KAKENHI 22500573, 216 25282216, 25560302, and 16H01869 to S.S."

4. Please ensure that you refer to Figure 3 in your text as, if accepted, production will need this reference to link the reader to the figure.

Reviewers' comments:

Reviewer's Responses to Questions

**Comments to the Author**

1. Is the manuscript technically sound, and do the data support the conclusions?

Reviewer #1: Partly

2. Has the statistical analysis been performed appropriately and rigorously? 

Reviewer #1: Yes

3. Have the authors made all data underlying the findings in their manuscript fully available?

Reviewer #1: Yes

4. Is the manuscript presented in an intelligible fashion and written in standard English?

Reviewer #1: Yes

5. Review Comments to the Author

Reviewer #1: The author tested the contract sensitivity of rats before and after intraperitoneal injection of serotonin re-uptake inhibitor in 2AFC-VDT behavior task. Contrast sensitivity was first found to be higher in rats after intraperitoneal injection of fluoxetine (FLX) than before in the condition on optimal special frequency of 0.1. The authors therefore concluded that endogenous 5-HT release in the brain can improve visual detectability. However, the function of FLX to inhibit 5-HTT is not directly related to the release of 5-HT, and FLX directly acts on the 5-HT receptor (Liang Peng et al Current Neuropharmacology 2014). It is better to improve the accuracy of conclusion in this study. Overall, this study suggests that FLX improves CS in rats which will provide more reference evidence for studying the relationship between 5-HT and visual information processing.

Some important points provided below might improve the accuracy of the conclusions.

1. Discuss the relationship between FLX, 5-HT and 5-HT receptors (Liang Peng et al. Current Neuropharmacology. 2014), which will help to clarify the relationship between 5-HT and visual information processing.

2. The conclusion in line 208 states that " serotonin endogenously released from the dorsal raphe nucleus...". There is no direct evidence in this article providing that 5-HT derived from DR functioning in CS , besides the major serotonergic neurons also exist in MR area (Iskra Pollak Dorocic et al. Neuron. 2014). Discussion of the serotonergic neurons projecting areas might help to improve the accuracy of the conclusion.

6. PLOS authors have the option to publish the peer review history of their article (what does this mean?). If published, this will include your full peer review and any attached files.

Reviewer #1: No

---

## [Author Response · Author response to Decision Letter 0]

12 Feb 2020

Professor Judith Homberg

Academic Editor

PLOS ONE

Re: PONE-D-19-31125

Serotonin improves behavioral contrast sensitivity of freely moving rats

PLOS ONE

Dear Professor Homberg,

Thank you very much for sending us your decision and the reviewer's comments on our manuscript, "Serotonin improves behavioral contrast sensitivity of freely moving rats" by Sato et al. We appreciate the positive comments and suggestions that have helped improve our study. Accordingly, we have revised our manuscript. In the process of the revision, we found that the population data of water intake and open field test contained one rat which was used in a preliminary experiment to determine their experimental conditions and whose contrast sensitivity was not measured. Therefore, we removed data of the rat from the population data of water intake and open field, by which the number of animals became equally seven in contrast sensitivity and water intake and open field. The statistical significance of the data was not changed after this treatment. Accordingly, the relevant part of text and Fig 4 have been modified.

Details of our responses to the reviewer’s comments are as follows.

Responses to Journal requirements

1. Please ensure that your manuscript meets PLOS ONE's style requirements, including those for file naming. The PLOS ONE style templates can be found at http://www.plosone.org/attachments/PLOSOne_formatting_sample_main_body.pdf and http://www.plosone.org/attachments/PLOSOne_formatting_sample_title_aut

hors_affiliations.pdf

We have confirmed The PLOS ONE style templates and modified our manuscript to meet the style requirements.

We have uploaded our data to figshare, and the DOI is as follows;

DOI: 10.6084/m9.figshare.11845611

"This work was supported by KAKENHI 22500573, 216 25282216, 25560302, and 16H01869 to S.S."

"The author(s) received no specific funding for this work.

We have removed the text related funding information in the Acknowledgments section of the revised version of our manuscript. We would like to update the financial disclosure section with the following funding information; “This work was supported by KAKENHI JP22500573, JP25282216, JP25560302, and JP16H01869 to S.S, and JP17J08499 to A.Y.S.”

4. Please ensure that you refer to Figure 3 in your text as, if accepted, production will need this reference to link the reader to the figure.

We have specified reference to Figure 3 on line 177 of the revised manuscript.

 

Responses to the Reviewer1

We would like to express our thanks to this reviewer for his insightful comments that have very much helped us improve the paper.

To the reviewer’s comments

1. Information about that FLX directly acts on the 5-HT receptor.

Reviewer 1 suggested discussing the activation 5-HT receptor by FLX. We have discussed this point in the Discussion section (line 236-242 in the manuscript).

2. Information about projection of dorsal raphe nucleus (DR) and median raphe nucleus (MR).

 Reviewer 1 suggested discussing serotonin endogenously released from not only dorsal but also median raphe nucleus. We have discussed this point in the Discussion section (line 224-228 in the manuscript).

Again, we would like to thank you, Prof. Judith Homberg, and the reviewers for your time and feedback. We hope the revised manuscript will be now considered suitable for publication in PLOS ONE.

With best wishes,

Satoshi Shimegi

---

## [Decision Letter · Decision Letter 1]

28 Feb 2020

Serotonin improves behavioral contrast sensitivity of freely moving rats

PONE-D-19-31125R1

Dear Dr. Shimegi,

We are pleased to inform you that your manuscript has been judged scientifically suitable for publication and will be formally accepted for publication once it complies with all outstanding technical requirements.

With kind regards,

Judith Homberg

Academic Editor

PLOS ONE

Additional Editor Comments (optional):

Reviewers' comments:

Reviewer's Responses to Questions

**Comments to the Author**

1. If the authors have adequately addressed your comments raised in a previous round of review and you feel that this manuscript is now acceptable for publication, you may indicate that here to bypass the “Comments to the Author” section, enter your conflict of interest statement in the “Confidential to Editor” section, and submit your "Accept" recommendation.

Reviewer #1: All comments have been addressed

2. Is the manuscript technically sound, and do the data support the conclusions?

Reviewer #1: Partly

3. Has the statistical analysis been performed appropriately and rigorously? 

Reviewer #1: Yes

4. Have the authors made all data underlying the findings in their manuscript fully available?

Reviewer #1: Yes

5. Is the manuscript presented in an intelligible fashion and written in standard English?

Reviewer #1: Yes

6. Review Comments to the Author

Reviewer #1: (No Response)

7. PLOS authors have the option to publish the peer review history of their article (what does this mean?). If published, this will include your full peer review and any attached files.

Reviewer #1: Yes: Chao Guo

---

## [Editor Report · Acceptance letter]

5 Mar 2020

PONE-D-19-31125R1 

Serotonin improves behavioral contrast sensitivity of freely moving rats 

Dear Dr. Shimegi:

I am pleased to inform you that your manuscript has been deemed suitable for publication in PLOS ONE. Congratulations! Your manuscript is now with our production department. 

With kind regards,

on behalf of

Dr. Judith Homberg 

Academic Editor

PLOS ONE